# AutoMoE: Neural Architecture Search for Efficient Sparsely Activated Transformers

## Abstract

Neural architecture search (NAS) has demonstrated promising results on identifying efficient Transformer architectures which outperform manually designed ones for natural language tasks like neural machine translation (NMT). Existing NAS methods operate on a space of dense architectures, where all of the sub-architecture weights are activated for every input. Motivated by the recent advances in sparsely activated models like the Mixture-of-Experts (MoE) model, we introduce sparse architectures with conditional computation into the NAS search space. Given this expressive search space which subsumes prior densely activated architectures, we develop a new framework `AutoMoE` to search for efficient sparsely activated sub-Transformers. `AutoMoE`-generated sparse models obtain (i) $4\times$ FLOPs reduction over manually designed dense Transformers and (ii) 23% FLOPs and 10% latency reduction over state-of-the-art NAS-generated dense sub-Transformers with parity in BLEU score on benchmark datasets for NMT. `AutoMoE` consists of three training phases: (a) Heterogeneous search space design with dense and sparsely activated Transformer modules (e.g., *how many experts? where to place them? what should be their sizes?*); (b) SuperNet training that jointly trains several subnetworks sampled from the large search space by weight-sharing; (c) Evolutionary search for the architecture with optimal trade-off between task performance and computational metrics like FLOPs and latency.

## 1 Introduction

Transformers have demonstrated state-of-the-art performance in several tasks, but the larger their size, the more difficult it is to use them in resource constrained settings (Strubell et al., 2019). Recent works in neural architecture search (NAS) (Wang et al., 2020; Xu et al., 2022a; 2021; So et al., 2021; Xu et al., 2022b; Javaheripi et al., 2022) have focused on identifying computationally efficient sub-Transformers that are easier to deploy on edge devices. However, existing works on NAS only focus on the subspace of dense[1] Transformer architectures, where all the network weights are activated for every input.

In contrast to the above dense models, sparsely activated ones like the Mixture-of-Experts (Fedus et al., 2022b) perform conditional computation in which only a subset of the weights of the network are activated per input. Selective compute allows us to design neural networks with a large number of model parameters, without significant increase in the computational cost. With increased capacity, these sparse models have demonstrated state-of-the-art performance in natural language tasks such as neural machine translation (NMT)(Kim et al., 2021; Kudugunta et al., 2021; Zuo et al., 2022).

*The goal of this work is to explore the space of sparsely activated MoE architectures for NAS to identify computationally efficient sparse sub-Transformers.*

Incorporating MoE architectures in the search space requires one to make several design choices. *(a) Expert placement:* Identifying the Transformer layers for introducing expert sub-networks. *(b) Number of experts:* How many experts to introduce in different layers? *(c) Expert FFN size*: What should be the feedforward network (FFN) size for each expert? Given the large search space of

---

[1]Terminologies: (1) **Dense architectures** refer to fully activated networks for every input. (2) **Sparse architectures** refer to sparsely activated ones with conditional computation per input. (3) **Optimal architectures** refer to Pareto-optimal ones with the best trade-off between task performance and computational metrics.

potential architectures and the exorbitant computational cost of training and evaluating them – existing approaches manually design MoE architectures with a highly-restricted homogeneous space. For instance, they use the same number of experts of the same capacity in different layers and make ad-hoc decisions like introducing experts in every other layer (Fedus et al., 2022b; Kim et al., 2021; Zuo et al., 2022; Du et al., 2022; Artetxe et al., 2021) or every four layers (Zoph et al., 2022).

These design choices are not necessarily optimal. The decoder should be lighter than the encoder for auto-regressive NMT tasks due to the cumulative latency of generating tokens one at a time (Liu et al., 2020; Kasai et al., 2021). This impacts the design choice for the number of decoder layers and the number of experts to use in each. For instance, the loss of capacity with decoder layer reduction can be compensated by adding experts on the remaining ones. On the encoder side, a vanilla placement of the maximum allowable experts in each layer results in increased latency from expert communication and activation, although theoretical FLOPs can remain unaffected. These suggest that the optimal MoE's could be heterogeneous when resources like latency or FLOPs are constrained. In a recent review on sparsely activated models, Fedus et al. (2022a) note that the optimal hyperparameters depend on application and resource specifications – where a systematic simulation of the compute, memory and communication cost can aid practitioners to quickly determine optimal settings without costly trial-and-error launches. `AutoMoE` provides such a framework to identify optimal hyper-parameter configurations for sparse models under computational constraints.

The above observations are depicted in Table 1, which shows demonstrative examples of manually designed architectures vs. those found by our `AutoMoE` framework from the search space. We compare these architectures against various computational metrics (e.g., latency, FLOPs, active MoE parameters), architectural configurations and task performance.

| Machine Translation | #Experts in each layer | | Accuracy | | Computational Footprint | |
|---|---|---|---|---|---|---|
| Design Approach | Encoder | Decoder | BLEU | Latency | # Active Params | FLOPs (G) |
| Manually designed (every layer) | 4-4-4-4-4-4 | 4-4-4-4-4-4 | 27.87 | 586ms | 56M | 3.4 |
| Manually designed (every other layer) | 1-4-1-4-1-4 | 1-4-1-4-1-4 | 28.48 | 506ms | 56M | 3.4 |
| `AutoMoE` (4 Experts) | 2-4-1-1-3-1 | 1-1-1-1-1 | 28.22 | 239ms | 49M | 3.1 |
| `AutoMoE` (4 Experts) | 1-1-4-4-4-1 | 4-1-1-1 | 28.15 | 194ms | 22M | 2.9 |

Table 1: Manually designed vs. `AutoMoE` searched architecture for 6-layer encoder-decoder Transformer. We report various computational footprint metrics (measured on 1 V100 GPU) and BLEU score of sparse expert models on WMT'14 En-De machine translation task. We show the number of experts per layer separated by hyphen (-) for encoder and decoder.

**Novelty:** To the best of our knowledge, AutoMoE introduces the first end-to-end framework to automatically design efficient MoE models under resource constraints. AutoMoE is also the first MoE framework to support *adaptive* computation due to heterogeneous experts, where input tokens are routed to experts of different sizes.

With this desiderata, we develop `AutoMoE` with the following components and contributions:

- We introduce a *heterogeneous search space* for Transformers consisting of variable number, FFN size and placement of experts in both encoders and decoders; variable number of layers, attention heads and intermediate FFN dimension of standard Transformer modules.

- We extend *Supernet training* to this new search space which combines all possible sparse architectures into a single graph and jointly trains them via weight-sharing, yielding a reduced amortized training cost.

- We use an evolutionary algorithm to *search for optimal sparse architecture* from Supernet with the best possible performance on a downstream task (e.g., BLEU score for NMT tasks) satisfying a user-specified computational constraint.

- Experiments on several NMT benchmarks demonstrate that `AutoMoE`-searched sparse models obtain (i) $4\times$ FLOPs reduction over manually designed dense Transformers and (ii) 23% FLOPs and 10% latency reduction over state-of-the-art NAS-generated dense sub-Transformers with comparable BLEU scores.

Changes made as part of the revision are highlighted in red.

## 2 BACKGROUND

**Sparse expert models:** Mixture-of-expert models have a rich literature in machine learning dating back to the early 90s (Yuksel et al., 2012). Sparsely activated expert models, where only a small subset of experts are active at any given time, have received significant attention with works such as Shazeer et al. (2017), Switch Transformers (Fedus et al., 2022b), GShard (Lepikhin et al., 2020), BASE (Lewis et al., 2021), Hash (Roller et al., 2021), GLaM (Du et al., 2022), Stochastic Experts (Zuo et al., 2022), Gating Dropout (Liu et al., 2022) and ST-MoE (Zoph et al., 2022). Some crucial differences in these works include choice of expert routing function, expert placement technique, stability/performance enhancement techniques and nature of the task (pre-training vs. fine-tuning). Some challenges in building sparse expert models include: (i) lack of diversity in expert design (expert layer selection, number of experts, expert size, etc.), (ii) training instability and (iii) expert load balancing issue, to name a few. A comprehensive review of recent sparse expert models can be found at Fedus et al. (2022a).

**Expert design:** Prior work on designing sparsely activated expert models has largely relied on ad-hoc manual choices in terms of expert layer selection, number of experts and their sizes. Existing approaches mostly use manual design, where they add experts on (i) alternate layers (Fedus et al., 2022b; Kim et al., 2021; Zuo et al., 2022; Du et al., 2022; Artetxe et al., 2021), (ii) every four layers (Zoph et al., 2022), or (iii) final few layers Rajbhandari et al. (2022). The resulting sparse models have homogeneous expert layers, i.e., same number of experts of the same size in all expert layers. These choices are generally agnostic to the computational constraints (e.g., latency, memory) of the hardware in which the sparse expert model has to be deployed.

**Switch Transformers:** The variant of sparse expert model used in this work is Switch Transformers (Fedus et al., 2022b), mainly chosen due to its popularity and high performance. Switch Transformers replace every Feed-Forward Network (FFN) layer with an expert layer consisting of a collection of experts (independent FFN networks). Each expert layer is preceded by a parameterized routing network that is trained to route each token to top-$k$ experts in the expert layer. In this work, we adapt Switch Transformers to an encoder-decoder model which is trained from scratch on the machine translation task using top-$1$ routing.

**Neural Architecture Search (NAS):** Given a search space of architectures and efficiency constraints (e.g., model size, latency), NAS typically aims to identify the optimal architecture that maximizes the task performance, while satisfying the efficiency constraints. The main challenges in building a NAS framework include: (i) constructing a search space that covers diverse architectures for the task, (ii) building a fast, accurate performance predictor and latency estimator for candidate architecture evaluation, and (iii) designing a search algorithm to find Pareto-optimal architectures. NAS has been recently used for natural language understanding tasks to build efficient BERT (Devlin et al., 2019) and GPT (Brown et al., 2020) based pre-trained language models (Xu et al., 2021; Yin et al., 2021; Xu et al., 2022a;b; Gao et al., 2022; Dong et al., 2021; So et al., 2021; Javaheripi et al., 2022) as well as for machine translation tasks (So et al., 2019; Wang et al., 2020). Hardware aware transformers (HAT) (Wang et al., 2020) is a state-of-the-art NAS framework for machine translation that uses hardware latency as feedback for optimization.

However, all of the above NAS works consider a search space with densely activated Transformer models, and primarily search over typical Transformer architectural hyper-parameters like number of layers, attention heads and hidden size. In contrast, we propose the first NAS framework that considers sparsely activated Transformer models like the Mixture-of-Experts, which subsume all prior densely activated Transformer models as a special case (i.e. one expert per layer). This allows us to expand the search space and make it more diverse by considering heterogeneous architectures and in the process address some longstanding design choices for MoE's like *how many experts? which layers to place the experts? what should be the expert size?* and so on.

## 3 NEURAL ARCHITECTURE SEARCH FOR MIXTURE-OF-EXPERTS

In this section, we present the components of the `AutoMoE` framework (as illustrated in Figure 1) for designing efficient sparse networks under computational constraints.

### 3.1 HETEROGENEOUS SEARCH SPACE

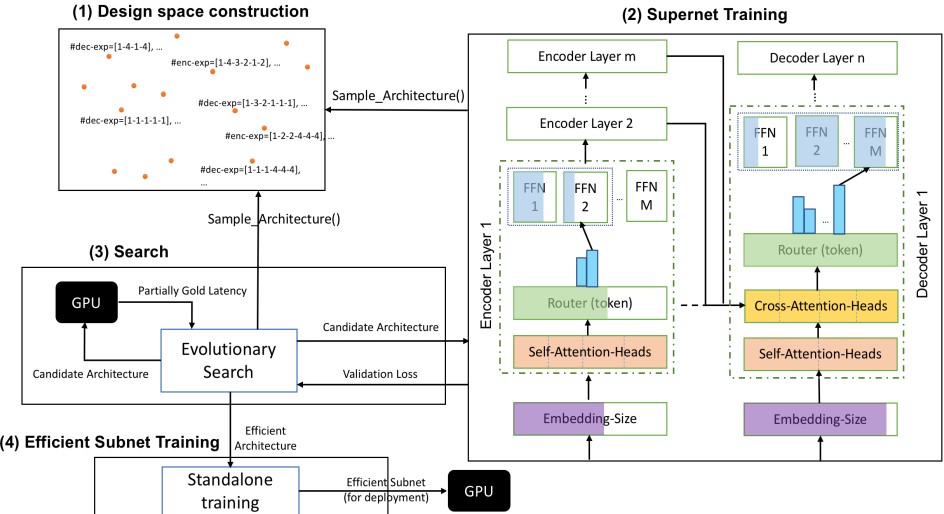

Figure 1: `AutoMoE` Framework. (1) Heterogeneous search space with variable dimensions for dense Transformer blocks and sparsely activated expert modules. (2) Supernet training by sampling subnetworks from search space and training them by sharing common weights with Supernet. (3) Evolutionary search to find efficient architectures by (a) sampling subnetworks from the search space; (b) using latency measured in the target device; and (c) performance estimation from Supernet as feedback for iterative optimization via crossover and mutation. (4) Efficient subnetwork(s) from evolutionary search is trained on downstream task.

Existing MoE approaches greatly restrict their design space by considering uniform distribution of capacity/size and number of expert subnetworks placed in different Transformer layers. For instance, the standard MoE design (Fedus et al., 2022b) for an $L$-layer Transformer with $M$ experts placed in alternate layers has only two possible configurations viz., $\{1\text{-}M\text{-}1\text{-}\cdots\}$, $\{M\text{-}1\text{-}M\text{-}\cdots\}$. Our design space allows *variable number of experts* in each layer resulting in $M^L$ possible configurations. (b) Furthermore, our design space also allows *variable expert capacity*, e.g., by modulating the width of the feedforward (FFN) subnetworks for different experts. Considering $N$ possible FFN dimensions for each expert results in $N^{M^L}$ possible configurations for designing the expert space. (c) Finally, given the autoregressive nature of tasks like neural machine translation, the inference cost is dominated by the decoder (Kasai et al., 2021). For instance, for token-based MoE, decoders take $200\times$ the time per

| Attributes | Dimensions |
|---|---|
| Encoder-Embedding-Size | $\{512, 640\}$ |
| Decoder-Embedding-Size | $\{512, 640\}$ |
| #Encoder-Layers | $\{6\}$ |
| #Decoder-Layers | $\{1, 2, 3, 4, 5, 6\}$ |
| Encoder-QKV-Dim | $\{512\}$ |
| Decoder-QKV-Dim | $\{512\}$ |
| #Encoder-Self-Att-Heads (PL) | $\{4, 8\}$ |
| #Decoder-Self-Att-Heads (PL) | $\{4, 8\}$ |
| #Decoder-Cross-Att-Heads (PL) | $\{4, 8\}$ |
| #Decoder-Arbitrary-Att (PL) | $\{-1, 1, 2\}$ |
| Encoder-FFN-Intermediate-Size (PL, PE) | $\{1024, 2048, 3072\}$ |
| Decoder-FFN-Intermediate-Size (PL, PE) | $\{1024, 2048, 3072\}$ |
| #Encoder-Experts (PL) | $\{1, 2, \cdots M\}$ |
| #Decoder-Experts (PL) | $\{1, 2, \cdots M\}$ |

Table 2: Search space of `AutoMoE`. 'PL' and 'PE' refer to per layer and per expert search dimensions. Decoder arbitrary attn. searches last $k$ encoder layers to attend. FFN size varies across layers and experts. $M$ denotes maximum experts per layer.

step compared to encoders at peak throughput (Kudugunta et al., 2021). Therefore, we further consider *variable number of decoder layers* along with the above choices for expert placement and expert capacity. **To the best of our knowledge, our work is the first to study such a flexible and exhaustive design space for modeling sparse architectures**.

In addition to the heterogeneous experts, we allow flexible design for the non-expert Transformer modules like the number of attention heads, hidden size and intermediate feedforward dimensions. This heterogeneous design of the non-expert, i.e., dense Transformer modules, has been explored in prior works such as HAT (Wang et al., 2020) for generation tasks like NMT, and AutoDistil (Xu et al., 2022a) for understanding tasks like those in the GLUE benchmark Wang et al. (2018).

A typical challenge in designing an expressive search space for NAS is the increased computational cost to search over all viable configurations. Some recent works like AutoDistil (Xu et al., 2022a) and few-shot NAS (Zhao et al., 2021) demonstrate that a curated search space can alleviate gradient conflicts in weight-sharing of candidate subnetworks and improve Supernet training and stability.

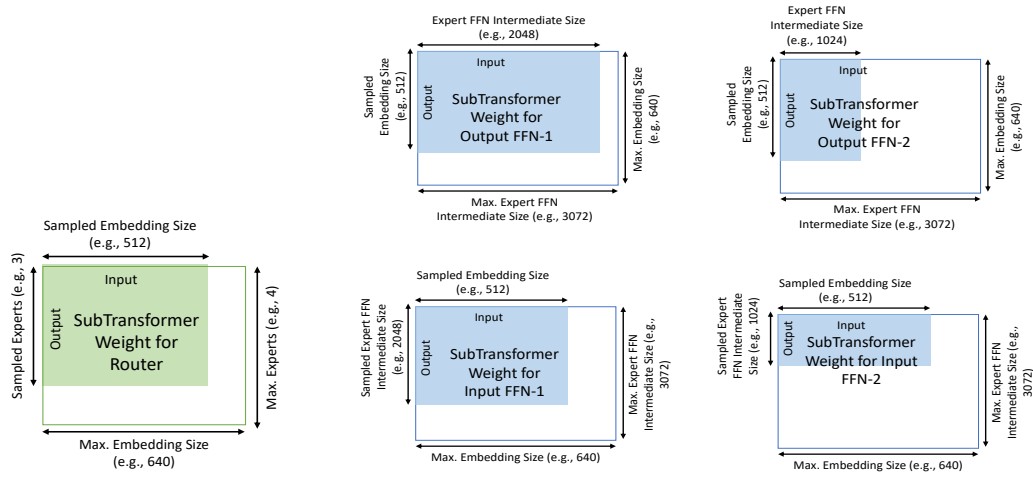

(a) Router         (b) Experts (e.g., 2 FFN experts)

Figure 2: Weight sharing in the Supernet for sparsely activated expert modules. Illustrations of weight sharing for the standard dense Transformer modules can be found at Wang et al. (2020).

Such conflicts result from sharing weights between subnetworks of very different sizes that have different convergence rates. To this end, we make the following design choices for our expert and non-expert Transformer modules. (i) For the *dense non-expert modules*, we leverage existing search space from prior work (Wang et al., 2020) which already demonstrated state-of-the-art performance for tasks like machine translation and forms the strongest baseline for our work in the dense Transformer space. (ii) We further incorporate various features for the *sparse expert modules* in the search space corresponding to the number of experts and their capacity in the form of their intermediate FFN size. We make the above choice for both encoders and decoders where each expert FFN can have a variable size and capacity. In both settings, each layer can have flexible choices for the standard Transformer hyper-parameters. (iii) Given the auto-regressive nature of machine translation tasks, the inference latency in the decoding phase can be twice as much as that in the encoding phase (Kasai et al., 2021) (demonstrated in Figure 3 (e)). Consequently, we also search over how many decoder layers to use and subsequent expert distribution based on latency constraint. We keep the number of encoder layers fixed for all our experiments and study this choice via ablation in Table 5. Table 2 shows our search space.

## 3.2 Supernet Training

A typical challenge in the NAS framework is to develop a performance estimator that can efficiently compute the accuracy of a candidate architecture. The naive approach of training candidate architectures from scratch to convergence and then evaluating on the validation set is prohibitively expensive given the large search space for all possible candidate architectures (see Section 3.3). To address this challenge, `AutoMoE`'s performance estimator is based on weight-sharing via a Supernet (Cai et al., 2020; Wang et al., 2020), which is a commonly used technique in recent NAS works. The Supernet for `AutoMoE` is the largest sparsely activated expert network in the search space. It consists of the maximum number of experts ($M$) placed in every layer of the Transformer in both encoder and decoder. Each expert FFN has the maximum intermediate hidden size in the search space. Similar principles apply to the non-expert dense modules initialized with corresponding full dimension.

The Supernet is trained with the following steps: (i) sample a candidate architecture randomly from the search space (Guo et al., 2020); (ii) train the sampled architecture by extracting the common portion of weights from different layers in the Supernet (i.e., by weight sharing) for one training step on the task; (iii) repeat steps (i) and (ii) until the training budget is exhausted. Once the Supernet training is complete, we can obtain a quick accuracy estimate for a candidate architecture (i.e. subnetwork) by extracting its shared weights from the Supernet and evaluating on the validation set.

The key challenge here is to build weight sharing techniques for expert-specific components, which include: (i) *router*: a neural network that is trained to route each token (of 'embedding size') in an incoming example to exactly one expert (out of $M$ experts) for top-1 routing; (ii) *FFN expert*: a standard Transformer FFN block that has unique weights and is learned independently. `AutoMoE`'s

expert layers follow the Switch Transformer (Fedus et al., 2022b) specification. For subnetwork extraction from the Supernet, `AutoMoE` extracts front rows and front columns of the Supernet's router weight matrix, corresponding to the subnet design. For example, consider the Supernet's router to be designed for 4 experts and 640 embedding size with the shape of the router weight matrix as $4 \times 640$. Consider a sampled subnet during Supernet training, consisting of $3 < 4$ experts and $512 < 640$ embedding size with the subnet's router matrix as $3 \times 512$. To populate this matrix, we extract the first 3 rows and first 512 columns from the Supernet's weight matrix (as illustrated in Figure 2 (a)). Such a weight sharing technique allows us to design architectures with varying number of experts in each Transformer layer.

`AutoMoE` also extracts front rows and front columns from the weight matrices of each FFN expert from the Supernet, corresponding to the subnet design. For the previous example, assume the intermediate FFN size of each expert in the Supernet to be 3072 (shape of weight matrix for first FFN layer is $3072 \times 640$ and second FFN layer is $640 \times 3072$). Assume the sampled subnet to be designed for 2 experts with intermediate FFN size of one expert to be 2048 while the other to be 1024. For the first expert, the weight matrices of the subnet of shape $2048 \times 512$ (Input) and $512 \times 2048$ (Output) are extracted from the first 2048 rows, 512 columns (Input) and first 512 rows, 2048 columns (Output) of the corresponding Supernet weights. For the second expert, the weight matrices of shape $1024 \times 512$ (Input) and $512 \times 1024$ (Output) are extracted from the first 1024 rows, 512 columns (Input) and first 512 rows, 1024 columns (Output) of the corresponding Supernet weights. This example is illustrated in Figure 2 (b). The subnet extraction technique does not extract weights from the third and fourth experts of the Supernet as the subnet is designed to have only two experts (not shown in the figure). Such a weight sharing technique allows us to design architectures with varying intermediate FFN size for each expert. Additional techniques for improving expert capacity such as stacking FFN layers, and techniques for improving Supernet performance with sandwich sampling (Yu et al., 2019), in-place knowledge distillation (Yu et al., 2019), gradient conflict reduction (Gong et al., 2022) are left for future work.

### 3.3 SEARCHING FOR EFFICIENT SUB-TRANSFORMER WITH COMPUTATIONAL CONSTRAINT

`AutoMoE` search is based on an evolutionary algorithm that takes the hardware computational constraint (e.g., latency $\leq$ 200ms) as input and aims to identify the sub-architecture which achieves maximum accuracy for the task while satisfying the constraint. The algorithm works by randomly sampling from the design space an initial set of sparsely activated candidate architectures; evolving the top architectures iteratively by mutation followed by crossover; until the search iterations are exhausted. Candidate architectures are easily ranked by the Supernet performance estimator based on the validation score for the task. Latency estimate for each architecture is obtained by measuring the latency directly on the target device. The standard approach measures gold latency for forward propagation of a batch of examples for a large number (e.g., 300) of passes and then computes the truncated mean (after removing bottom and top 10% outlier latencies). This latency estimation can be costly given the large space of candidate architectures. To overcome this challenge, `AutoMoE` uses *partially gold latency*, which is obtained by forward propagation of a batch of examples for a small number (e.g., 100) of passes and then computing truncated mean. After the search is completed, the architecture with the highest performance is selected as the optimal one.

### 3.4 TRAINING EFFICIENT SUB-TRANSFORMER

Once the optimal sparsely activated architecture is identified, we train the model weights for the final architecture from scratch to convergence for the same number of training steps – following the evaluation setting of HAT (Wang et al., 2020) for a fair comparison.

## 4 EXPERIMENTS

**Datasets and evaluation metrics.**

We evaluate `AutoMoE` on standard machine translation benchmarks: WMT'14 En-De, WMT'14 En-Fr and WMT'19 En-De with dataset statistics in Table 10. We use pre-processed datasets and evaluation setup from Wang et al. (2020). We report BLEU score (Papineni et al., 2002) as a performance metric with beam of size 5 and a length penalty of 0.6 (for WMT).

**Baselines and `AutoMoE` variations.** We compare `AutoMoE` against both manually designed and NAS-searched architectures. For the manual baseline, we consider: (a) densely activated Transformers Vaswani et al. (2017a) with no experts; (b) sparsely activated MoE with homogeneous experts

| Dataset | Network | #Active Params (M) | Sparsity (%) | FLOPs (G) | BLEU | GPU hours |
|---------|---------|--------------------|--------------|-----------|------|-----------|
| **WMT'14 En-De** | | | | | | |
| Transformer | Dense | 176 | 0 | 10.6 | 28.4 | 184 |
| Transformer | Sparse (every layer) | 71 | 39 | 4.3 | **28.7** | - |
| Evolved Transformer | NAS over Dense | 47 | 0 | **2.9** | 28.2 | 2,192,000 |
| HAT | NAS over Dense | 56 | 0 | 3.5 | 28.2 | 264 |
| Random Search | NAS over Sparse | 27 | 50 | 2.2 | 27.3 | 126 |
| AutoMoE (6 Experts) | NAS over Sparse | **45** | 62 | **2.9** | 28.2 | 224 |
| **WMT'14 En-Fr** | | | | | | |
| Transformer | Dense | 176 | 0 | 10.6 | 41.2 | 240 |
| Transformer | Sparse (every layer) | 71 | 39 | 4.3 | 41.5 | - |
| Evolved Transformer | NAS over Dense | 175 | 0 | 10.8 | 41.3 | 2,192,000 |
| HAT | NAS over Dense | 57 | 0 | 3.6 | 41.5 | 248 |
| Random Search | NAS over Sparse | 27 | 50 | 2.2 | 40.3 | 130 |
| AutoMoE (6 Experts) | NAS over Sparse | **46** | 72 | **2.9** | 41.6 | 236 |
| AutoMoE (16 Experts) | NAS over Sparse | 135 | 65 | 3.0 | **41.9** | |
| **WMT'19 En-De** | | | | | | |
| Transformer | Dense | 176 | 0 | 10.6 | 46.1 | 184 |
| Transformer | Sparse (every layer) | 71 | 39 | 4.3 | **46.3** | - |
| HAT | NAS over Dense | 63 | 0 | 4.1 | 45.8 | 264 |
| Random Search | NAS over Sparse | 27 | 50 | 2.2 | 43.7 | 126 |
| AutoMoE (2 Experts) | NAS over Sparse | **45** | 41 | **2.8** | 45.5 | 248 |
| AutoMoE (16 Experts) | NAS over Sparse | 69 | 81 | 3.2 | 45.9 | |

Table 3: FLOPs improvement of AutoMoE vs. baselines with Pareto-optimal architectures highlighted in blue. Changes in revision are highlighted in red. We report both active model parameters; and sparsity measured as non-active parameters as a percentage of total parameters. We report training time for one Nvidia V100 GPU and training cost following HAT (Wang et al., 2020) corresponding to $40K$ training steps.

(i.e. same number and FFN size) placed in every other layer (Fedus et al., 2022b; Kim et al., 2021; Zuo et al., 2022; Du et al., 2022; Artetxe et al., 2021). For the NAS baselines, we consider (c) HAT (Wang et al., 2020), which is a Supernet-based state-of-the-art NAS framework for identifying efficient densely activated sub-Transformers for neural machine translation (same task setting as ours); and (d) Evolved Transformer (So et al., 2019) which is one of the earlier works on finding efficient dense sub-Transformers with evolution-based architecture search.

**Training configurations.** All the baselines and `AutoMoE` variants including the Supernet and final model are trained with the same setting for fair comparison, following HAT. All the models are trained for $40K$ steps, with a warmup of $10K$ steps from $10^{-7}$ to $10^{-3}$ and use cosine annealing to $10^{-7}$ for the rest of the steps. All models are trained using fairseq toolkit (Ott et al., 2019) with an effective batch size of $524K$ tokens on 16 V100 GPUs.

**Evolutionary search setup.** For performance estimation, we monitor the validation loss of the subnet on the NMT task. We compute latency by measuring the time taken to perform translation from a source sentence to a target sentence with same desired input/output length (30 for WMT) and original beam settings (see Section 4) on the target device (NVIDIA V100 GPU). We measure latency 300 times for gold (to report final metrics) and 100 times for partially gold (during evolutionary search) respectively; discard the top and bottom 10% (outlier latency) and compute mean of the rest. Hyper-parameter settings for evolutionary search include: 15 as iterations, 125 as population size, 25 as parents' size, 50 as mutation population size with mutation probability of 0.3 and 50 as crossover population size. Unless otherwise stated, the latency constraint for all the experiments is set to 200 ms.

## 5 RESULTS

### 5.1 AUTOMOE VS. BASELINES – PERFORMANCE COMPARISON

Table 3 presents the comparison of `AutoMoE` with baseline models on several computational metrics and task performance. We report the number of parameters without embedding weights, and FLOPs without the last decoding layer for all the models, consistent with Wang et al. (2020) evaluation settings. `AutoMoE`-generated sparsely activated sub-Transformers obtain $4\times$ reduction in FLOPs over manually designed (densely-activated) Transformer-big. Compared to NAS baselines

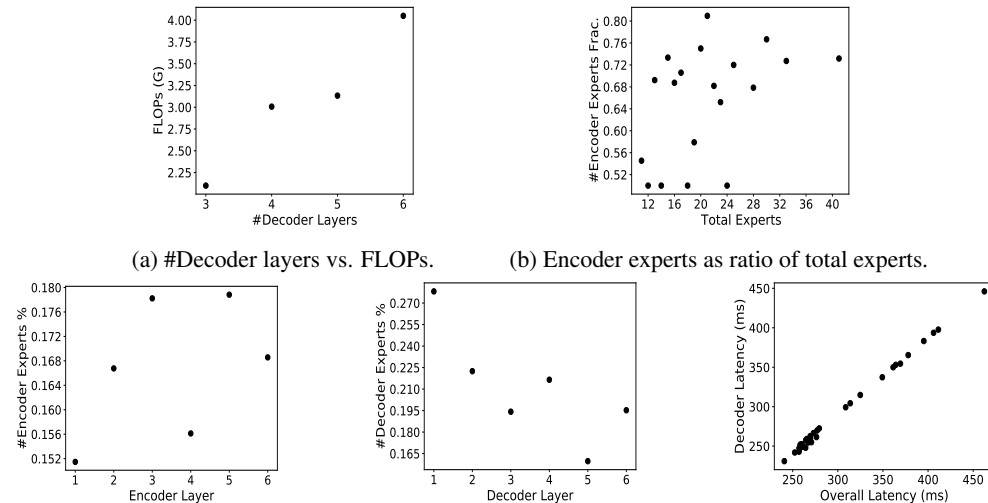

(a) #Decoder layers vs. FLOPs.     (b) Encoder experts as ratio of total experts.

(c) Encoder layer vs. #Expert (%).    (d) Decoder layer vs. #Expert (%).    (e) Overall vs. Decoder latency.

Figure 3: Architecture analysis for `AutoMoE`-generated sparsely activated models. We sample several architectures from the Pareto frontier for `AutoMoE`-variants and baselines, and report the aggregate statistics in terms of the impact on different computational metrics.

like Evolved Transformer So et al. (2019) and HAT Wang et al. (2020) that generate densely activated sub-Transformers, `AutoMoE` improves on both FLOPs and BLEU score on an aggregate across all tasks. Notably, Supernet-based `AutoMoE` and HAT have a massively reduced amortized training cost (GPU hours) compared to Evolved Transformer with progressive evolutionary search.

Compared to all the other densely-activated models, we observe that `AutoMoE` generates networks with high sparsity resulting in massively reduced active parameters and active model size. For the NAS models, we train the top-2 sub-Transformers in the Pareto frontier and report the one with the best BLEU vs. FLOPs trade-off on the validation set. The optimal maximum experts vary for different tasks, with 6 experts for WMT'14 En-De and 16 experts for WMT'14 En-Fr and WMT'19 En-De – given that the latter two datasets are $10\times$ larger than the former.

| Dataset | FLOPs | Latency (ms) | BLEU |
|---|---|---|---|
| **WMT'14 En-De** | | | |
| HAT | 3.5 | 205 | 28.2 |
| AutoMoE | 2.9 | 190 | 28.2 |
| **WMT'14 En-Fr** | | | |
| HAT | 3.6 | 212 | 41.5 |
| AutoMoE | 2.9 | 196 | 41.6 |
| **WMT'19 En-De** | | | |
| HAT | 4.1 | 212 | 45.8 |
| AutoMoE | 2.8 | 180 | 45.5 |

Table 4: AutoMoE reduces FLOPs by 23% and latency by 10% over optimal HAT (Wang et al., 2020) architectures under same latency constraint (200 ms) in aggregate with similar task performance – demonstrating NAS over sparse networks is more efficient than NAS over dense networks.

## 5.2 ANALYSIS

**Decoder layers vs. FLOPs.** Figure 3 (a) shows the average FLOPs for all `AutoMoE` variants with different decoder layers as obtained during our search (varying from 3 to 6) and baseline models. Notice that the FLOPs increase with increase in decoder layers, given the auto-regressive nature of NMT tasks which require generating tokens sequentially. In contrast to manually designed Transformers with 6 decoder layers (both dense and sparsely activated MoE variants), `AutoMoE`- and HAT-searched architectures reduce the number of decoder layers with a resulting decrease in both FLOPs and latency. This is also evident in Figure 3 (e) which shows that decoder latency dominates the total inference latency for all the models by more than 90%.

**Expert distribution in encoder vs. decoder.** Figure 3 (b) plots the number of encoder experts as ratio of total experts for `AutoMoE`-generated sub-Transformers. We observe that `AutoMoE` assigns a significant number of experts to the encoder as compared to the decoder. As a result, encoders have much higher capacity (i.e., encoder parameters as a proportion of overall parameters) than decoders. This correlates with the earlier observation that models with higher encoder layers compared to decoder layers enjoy better latency-performance trade-off (Kasai et al., 2021). Our findings from

`AutoMoE` designed architectures indicate that the number of layers and experts are two knobs that jointly help in modulating encoder capacity and decoder latency to design efficient models.

**Expert distribution in different layers.** Figures 3 (c) and (d) show the percentage of experts allocated to different layers for encoders and decoders – averaged over several sampled architectures. Notice that the middle encoder layers ($3^{rd}$, $5^{th}$) are allocated the maximum number of experts, while the first layer receives the least. The trend reverses for decoder, with the first layer receiving most experts with gradual reduction in expert allocation. This is also consistent with keeping decoders light by dropping layers to reduce latency, while compensating for the reduced capacity with increased experts in the first few layers.

**Pareto-optimal `AutoMoE` designed architectures.** Table 7 in Appendix shows the sparsely activated expert architectures designed by two variants of `AutoMoE` ('std-expert': expert FFN size same in each layer and variable across; 'fract-expert': fully heterogeneous expert size) for different datasets with the best trade-off in BLEU vs. latency. On aggregate 69% of the experts are allocated to the encoder compared to the decoder. Meanwhile, 70% of the expert layers in 'fract-expert' architectures have 2 or more experts, out of which more than 75% of the expert layers have varying capacities (i.e., experts with different FFN intermediate size).

**Search space variations.** Table 5 presents the impact of different search space choices on the efficiency and performance trade-off. The first variation is to make '#Encoder Layers' an elastic search dimension. Note that both HAT and `AutoMoE` consider the number of encoder layers to be fixed (refer to Table 2). We observe that varying encoder layers degrades the model efficiency in terms of FLOPs (top major row), re-inforcing our prior observations on the importance of encoder capacity and depth.

In the second variation (Table 5, second major row), we fix the expert architecture (with 2 experts manually placed uniformly) in the search space and only search for the standard Transformer hyper-parameters. Observe that `AutoMoE`-designed experts have better FLOPs than such manually designed MoE architectures.

| Search Space Variation | BLEU | FLOPs (G) |
|---|---|---|
| **Varying number of encoder layers** | | |
| HAT | 28.2 | 3.5 |
| HAT w/ #Encoder-Layers $\in \{1-6\}$ | 28.1 | 3.4 |
| AutoMoE (Std-expert, 2 Experts) w/ fixed encoder layers | 28.2 | 2.9 |
| AutoMoE (Std-expert, 2 Experts) w/ #Encoder-Layers $\in \{1-6\}$ | 28.3 | 3.7 |
| AutoMoE | 28.2 | 2.9 |
| **AutoMoE (Std-expert, 2 Experts) w/ manually designed homogeneous experts** | | |
| 1-2-1-2-1-2 | 28.3 | 3.5 |
| 1-1-1-2-2-2 | 28.3 | 3.8 |
| 2-2-2-1-1-1 | 28.3 | 3.1 |
| **AutoMoE (Fract-expert, 2 experts)** | | |
| AutoMoE (Fract-expert) | 28.4 | 3.5 |
| w/ Identity Expert - FFN Intermediate size $\in \{0, 3072\}$ | 28.1 | 2.7 |

Table 5: Variations in `AutoMoE`'s search space on WMT'14 En-De dataset.

The last variation introduces identity or dummy experts (i.e., expert with 0 intermediate FFN size, equivalent to identity operation). This explores the idea that we can *skip* the computation for some of the tokens based on context rather than always forcing them through an FFN. We observe that identity experts marginally hurt the performance but significantly reduce FLOPs (last major row).

# 6 CONCLUSION

`AutoMoE` is the *first* framework to explore the space of sparsely activated Mixture-of-Experts (MoE) models for neural architecture search (NAS). `AutoMoE` identifies efficient sparsely-activated sub-Transformers with reduction in FLOPs and latency over both manually designed and NAS-searched architectures, with parity in BLEU score on benchmark machine translation (MT) tasks. `AutoMoE` explores a fully heterogeneous search space with variable number of experts, their size and placement choices in different layers for encoders and decoders, alongside other standard Transformer architectural hyper-parameters. Our experiments show that AutoMoE reduces FLOPs by 23% and latency by 10% over optimal HAT architectures under the same latency constraint in the aggregate, across benchmark MT tasks, while maintaining similar task performance.

Given our focus on finding efficient MoE models *under computational constraints*, `AutoMoE` search space and evaluation has been restricted in scale to big-sized Transformer models for benchmark MT tasks. A natural extension of this work is to explore the limits of MoE models like SwitchTransformers (Fedus et al., 2022b) and GShard (Lepikhin et al., 2020) that are significantly larger containing billions to trillions of parameters; as well as designing sparse and transferable efficient expert models (Zoph et al., 2022) for diverse types of tasks like reasoning, summarization and understanding.

## 7 REPRODUCIBILITY STATEMENT

The source code to reproduce `AutoMoE` results can be found as part of the supplementary material. There will be a 'readme.md' file that has scripts to run the complete pipeline:

1. Download WMT datasets from web.
2. Train Supernet for performance estimation (Section 3.2).
3. Run evolutionary search to identify best subnet (Section 3.3).
4. Compute latency of best subnet (Section 4).
5. Compute FLOPs of best subnet (Section 4).
6. Train best subnet from scratch (Section 3.4).
7. Compute BLEU score of best subnet.

Training configurations of supernet, evolutionary search hyperparameters and training configurations of subnet has been discussed in Section 4.

`AutoMoE` is built over fairseq (Ott et al., 2019) and HAT (Wang et al., 2020) implementation (publicly available at `https://github.com/mit-han-lab/hardware-aware-transformers`). For fair comparison, we retain most of the hyper-parameters, training and evaluation recipes, preprocessed datasets from the original HAT implementation.

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

# A APPENDIX

| Hyperparameter | HAT | AutoMoE (std-experts) | AutoMoE (fract-experts) | Trans-Base / Big |
|---|---|---|---|---|
| Encoder-Embedding-Size | {512, 640} | {512, 640} | {512, 640} | 512 / 1024 |
| Decoder-Embedding-Size | {512, 640} | {512, 640} | {512, 640} | 512 / 1024 |
| #Encoder-Layers | {6} | {6} | {6} | 6 |
| #Decoder-Layers | {1, 2, 3, 4, 5, 6} | {1, 2, 3, 4, 5, 6} | {1, 2, 3, 4, 5, 6} | 6 |
| Encoder-QKV-Dim | {512} | {512} | {512} | 512 / 1024 |
| Decoder-QKV-Dim | {512} | {512} | {512} | 512 / 1024 |
| #Encoder-Self-Att-Heads (PL) | {4, 8} | {4, 8} | {4, 8} | 8 / 16 |
| #Decoder-Self-Att-Heads (PL) | {4, 8} | {4, 8} | {4, 8} | 8 / 16 |
| #Decoder-Cross-Att-Heads (PL) | {4, 8} | {4, 8} | {4, 8} | 8 / 16 |
| #Decoder-Arbitrary-Att (PL) | {-1, 1, 2} | {-1, 1, 2} | {-1, 1, 2} | -1 |
| Encoder-FFN-Intermediate-Size (PL) | {1024, 2048, 3072} | {1024, 2048, 3072} | {1024, 2048, 3072} | 2048 / 4096 |
| Decoder-FFN-Intermediate-Size (PL) | {1024, 2048, 3072} | {1024, 2048, 3072} | {1024, 2048, 3072} | 2048 / 4096 |
| #Encoder-Experts (PL) | - | {1, 2, $\cdots$ M} | {1, 2, $\cdots$ M} | - |
| #Decoder-Experts (PL) | - | {1, 2, $\cdots$ M} | {1, 2, $\cdots$ M} | - |
| Enc-Expert-FFN-Inter-Size (PL, PE) | - | - | {1024, 2048, 3072} | - |
| Dec-Expert-FFN-Inter-Size (PL, PE) | - | - | {1024, 2048, 3072} | - |

Table 6: Search space of AutoMoE vs. HAT vs. manually designed Transformer-Base / Big (Vaswani et al., 2017b). 'PL' refers to per layer while 'PE' refers to per expert. $M$ denotes maximum number of experts per layer.

**Qualitative analysis of expert architectures.** Table 7 presents the expert architectures designed by both variants of AutoMoE across datasets. On an average, more than 69% of the experts get assigned to the encoder compared to the decoder. As a result, overall capacity of the encoder can be significantly increased by introducing more experts, without having a drastic impact on the overall model latency and FLOPs. On an average, more than 70% of the expert layers in 'fract-expert' architectures have 2 or more experts, out of which more than 75% of the expert layers have varying capacities (i.e., expert with different FFN intermediate size).

| Model Dataset | Encoder | | Decoder | |
|---|---|---|---|---|
| | #Experts per layer | Expert FFN Inter Size | #Experts per layer | Expert FFN Inter Size |
| **Std-expert** | | | | |
| WMT'14 En-De | 3-2-4-1-5-2 | 2048-3072-3072-3072-3072-3072 | 2-1-1-1 | 2048-3072-3072-3072 |
| WMT'14 En-Fr | 1-1-3-1-3-3 | 3072-2048-3072-3072-3072-3072 | 2-1-1-1 | 3072-3072-3072-3072 |
| WMT'19 En-De | 1-2-5-1-2-4 | 3072-3072-3072-3072-3072-3072 | 4-4-1-1 | 3072-3072-3072-3072 |
| **Fract-expert** | | | | |
| WMT'14 En-De | 3-2-3-4-1-3 | [2048-3072-2048]-[3072-1024]-[3072-3072-1024]-[3072-1024-3072-2048]-3072-[3072-1024-3072] | 3-1-1-1 | [3072-1024-2048]-3072-3072-3072 |
| WMT'14 En-Fr | 6-2-3-4-4-5 | [2048-1024-2048-1024-1024-3072]-[2048-2048]-[3072-3072-2048]-[3072-3072-2048-3072]-[3072-1024-1024-2048]-[2048-3072-3072-2048-2048] | 2-1-4-2 | [3072-3072]-3072-[3072-3072-3072-2048]-[3072-2048] |
| WMT'19 En-De | 2-3-1-2-6-1 | [3072-3072]-[3072-3072-3072]-3072-[3072-2048]-[3072-1024-2048-3072-1024-2048]-3072 | 2-4-1-1 | [3072-3072]-[3072-1024-2048-3072]-3072-3072 |

Table 7: AutoMoE-generated Pareto-optimal architectures for different datasets. FFN intermediate sizes for fractional experts (i.e. varying expert sizes within each layer) are enclosed within square brackets.

**Full Architecture Design.** Figure 4, 5 and 6 present the full architecture design of pareto-efficient architectures generated by AutoMoE.

**Partially Gold Latency vs. Latency Predictor.** Table 8 shows the comparison of different latency estimators: proposed partially gold latency and latency predictor. On all datasets, our proposed par-

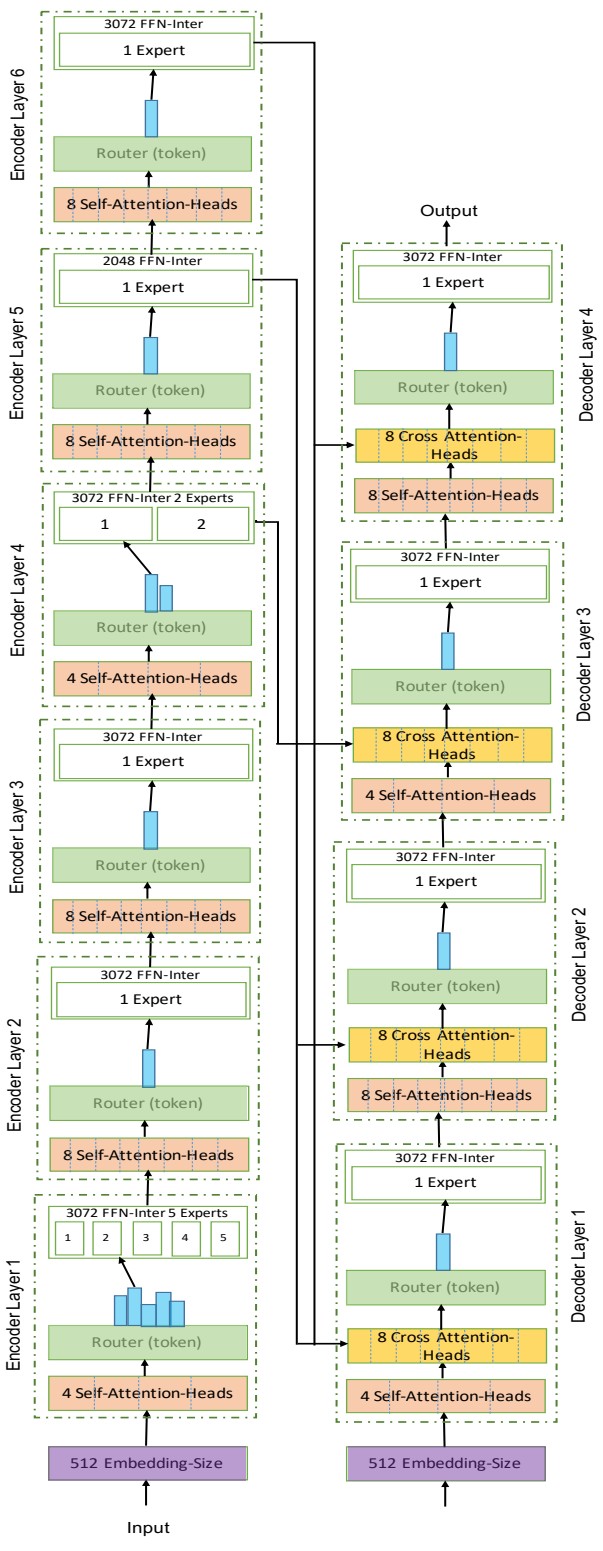

Figure 4: `AutoMoE`-generated architecture for WMT'14 En-De.

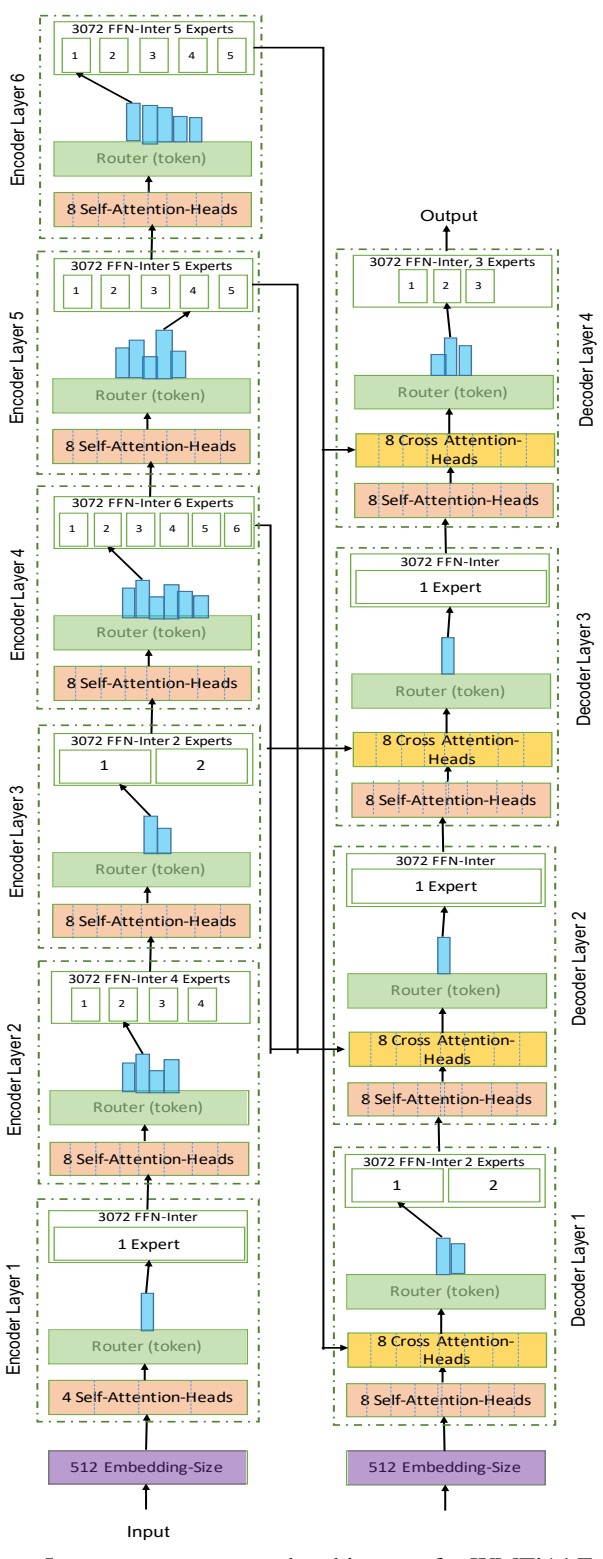

Figure 5: `AutoMoE`-generated architecture for WMT'14 En-Fr.

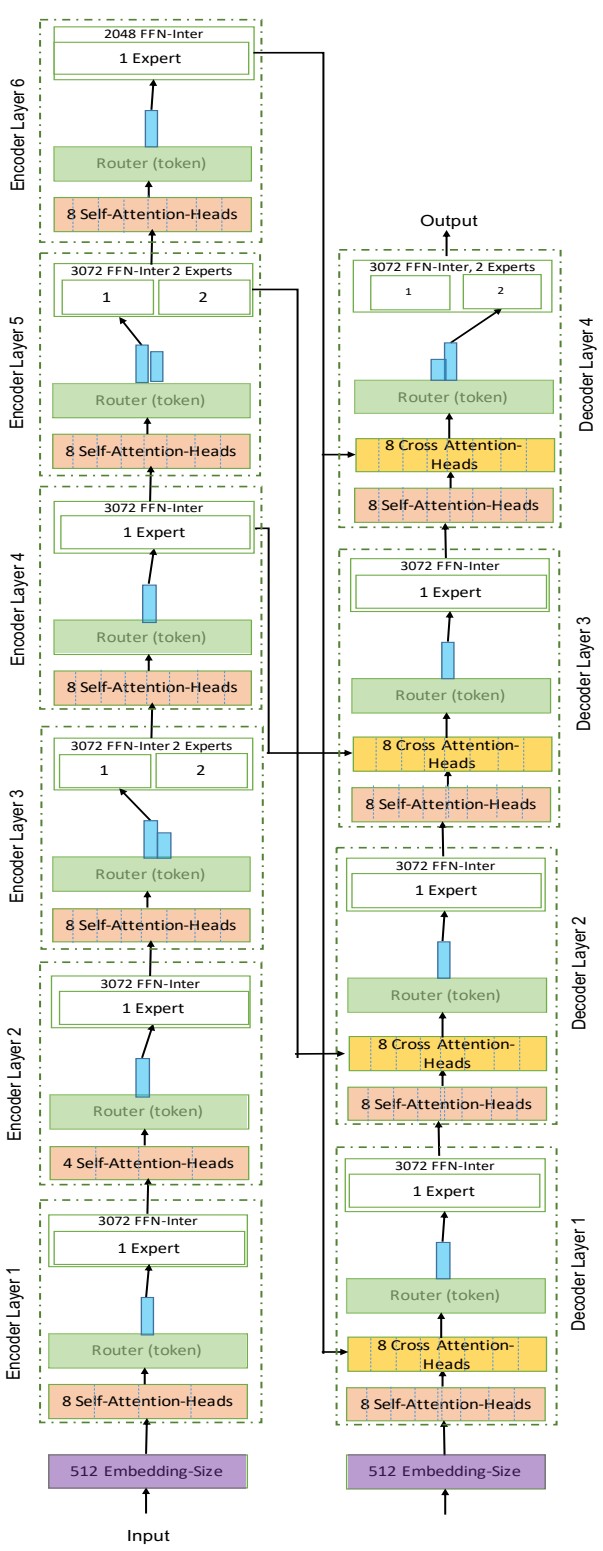

Figure 6: `AutoMoE`-generated architecture for WMT'19 En-De.

| Dataset | Latency Estimator | #Active Params (M) | Sparsity (%) | FLOPs (G) | BLEU | Latency (ms) |
|---|---|---|---|---|---|---|
| **WMT'14 En-De** | | | | | | |
| AutoMoE (2 Experts) | Partially Gold Latency | 46 | 29 | 2.9 | 28.1 | 178 |
| AutoMoE (2 Experts) | Latency Predictor | 46 | 29 | 2.9 | 28.2 | 189 |
| **WMT'14 En-Fr** | | | | | | |
| AutoMoE (2 Experts) | Partially Gold Latency | 46 | 29 | 2.9 | 41.2 | 176 |
| AutoMoE (2 Experts) | Latency Predictor | 46 | 29 | 2.9 | 41.1 | 190 |
| **WMT'19 En-De** | | | | | | |
| AutoMoE (2 Experts) | Partially Gold Latency | 45 | 40 | 2.8 | 45.5 | 180 |
| AutoMoE (2 Experts) | Latency Predictor | 51 | 40 | 3.2 | 46.0 | 229 |

Table 8: Latency improvement with the proposed partially gold latency vs. latency predictor. We report both active model parameters; and sparsity measured as non-active parameters as a percentage of total parameters.

| Search Constraint | BLEU | FLOPs (G) | Latency (ms) |
|---|---|---|---|
| **Latency ≤ 200ms** | | | |
| `AutoMoE` (2 Experts) | 41.23 | 2.9 | 176 |
| `AutoMoE` (4 Experts) | 41.22 | 3.0 | 198 |
| **FLOPs ≤ 3 GFLOPs** | | | |
| `AutoMoE` (2 Experts) | 41.09 | 3.0 | 216 |
| `AutoMoE` (4 Experts) | 41.10 | 3.0 | 229 |

Table 9: Impact of different search constraints on WMT'14 En-Fr dataset.

tially gold latency yields better latency, for same or better FLOPs, BLEU, active model parameters and sparsity.

**Latency vs. FLOPs as constraint for search.** Table 9 presents the impact of latency and FLOPs as computational constraints on the performance-efficiency trade-off. Constraining FLOPs results in models that fully exhaust the FLOPs budget for 3 GFLOPs and 4 GFLOPs; while leading to higher latency. On the other hand, constraining the latency tends to underutilize the budget and leads to relatively superior FLOPs and latency, thereby providing a stricter control.

| Dataset | Year | Source Lang | Target Lang | #Train | #Valid | #Test |
|---|---|---|---|---|---|---|
| WMT | 2014 | English (en) | German (de) | 4.5M | 3000 | 3000 |
| WMT | 2019 | English (en) | German (de) | 43M | 2900 | 2900 |
| WMT | 2014 | English (en) | French (fr) | 35M | 26000 | 26000 |

Table 10: Machine translation benchmark data.

