# OpenReview forum: "AutoMoE: Neural Architecture Search for Efficient Sparsely Activated Transformers"
_ICLR.cc/2023/Conference — Submitted to ICLR 2023_

### Official Review · Reviewer_CMSP · 2022-10-22

**Confidence:** 4
**Clarity, Quality, Novelty And Reproducibility:** I think this paper is well-written an…
**Correctness:** 3
**Technical Novelty And Significance:** 2
**Empirical Novelty And Significance:** 2
**Recommendation:** 5

**Strength And Weaknesses:**

Strength:
1. This paper is well-written.
2. The practical improvements over previous methods are clear, and the search cost is affordable.

Weakness:
1. 1. I feel this work is a straightforward application of existing NAS methods on MoE with some engineering efforts. The novelty of the proposed framework is limited. According to the experiments, the improvements are not surprising, given that the search space and the overall framework are much more complicated than previous methods.
2. In addition to FLOPs, the GPU latency of the models should be included in Table 4.


**Summary Of The Paper:**

This paper introduces AutoMoE, an AutoML framework for searching efficient sparsely activated models. Experiments are conducted on three machine translation benchmark datasets, including WMT'14 En-De, WMT'14 En-Fr, and WMT'19 En-De. Compared with previous models (e.g., HAT), AutoMoE achieves a better trade-off between FLOPs and BLEU scores.

**Summary Of The Review:**

I have some concerns regarding the novelty of this work. And it is unclear to me whether AutoMoE is better than previous models when comparing their efficiency on hardware (e.g., GPU).

---

> ### Author Response · Authors · 2022-11-12
> **Response to reviewer CMSP**
>
> Thank you for your time to review our work and provide feedback. We respond to quoted comments below:
> * *“I feel this work is a straightforward application of existing NAS methods on MoE with some engineering efforts. The novelty of the proposed framework is limited. ”*
>
> We **respectfully disagree** with this comment. See General Response, (1) and (2) for an explanation of the novelty of our work. See General Response, (3)  for the major differences between HAT and AutoMoE.
> **To summarize:**
>    1. AutoMoE is the first work on NAS for MoE.
>    2. Introduces the notion of adaptive computation with heterogeneous experts that is not explored in prior MoE works. This shortcoming has been posed as a future research direction in several recent works (MoE survey - Fedus et al., 2022; ST-MoE - Zoph et al., 2022).
>    3. AutoMoE incorporates MoE search space over HAT (which adopted the same search space as Transformers). This requires non-trivial sampling and routing designs (Figure 2); latency evaluators due to the stochastic nature of routing (Section 3.3)
>    4. Scalability as our MoE SuperNets are E times larger than HAT SuperNet (e.g., E=16 for 16 experts)
>    5. Significant speedup over HAT (23% FLOPs and 10% latency reduction over optimal HAT architectures)
>
> * *“According to the experiments, the improvements are not surprising, given that the search space and the overall framework are much more complicated than previous methods”*
>
> AutoME follows a similar trend as a lot of existing NAS research which improves upon prior work by enriching the search space and exploring new applications. For instance,
>    1. HAT (Wang et al., 2020) builds on standard Transformer search space. It did not introduce SuperNet training or evolutionary search. Instead, it demonstrated the end-to-end framework to work well for NMT tasks
>    2. PRIMER (So et al., NeurIPS 2022) enriches the standard decoder-only Transformer design space with a search space over low-level Tensorflow instructions
>    3. As another example, AutoBERT-Zero (Gao et al., 2022) enriches (i) the standard BERT design space with a pool of math operations, (ii) the standard evolutionary search with operation-priority evolutionary search
>    4. Similarly, AutoMoE bring the MoE search space into NAS and devises sampling and routing mechanisms to support the new search space
>    5. While SuperNet training and evolutionary search both are common to most of the above works — each one has made significant adaptations of standard training protocols to support their new search space.
>
> * “In addition to FLOPs, the GPU latency of the models should be included in Table 4.”
>
> AutoMoE generated architectures obtain 23% FLOPs and 10% latency reduction (while achieving a comparable performance) as shown in the following tables:
>
> **WMT’14 En-De**:
> |  Framework  | FLOPs (G) | Latency (ms) | BLEU |
> | ------------- |:-------------:|:-------------:|:-------------:|
> | HAT     | 3.5 | 205 | 28.2 |
> | AutoMoE |  2.9 | 190 | 28.2 |
> | | 17% speedup | 7.9% speedup |
>
> **WMT’14 En-Fr**:
> |  Framework  | FLOPs (G) | Latency (ms) | BLEU |
> | ------------- |:-------------:|:-------------:|:-------------:|
> | HAT     | 3.6 | 212 | 41.5 |
> | AutoMoE |  2.9 | 196 | 41.6 |
> | | 19% speedup | 7.5% speedup | |
>
> **WMT’19 En-De**:
> |  Framework  | FLOPs (G) | Latency (ms) | BLEU |
> | ------------- |:-------------:|:-------------:|:-------------:|
> | HAT     | 4.1 | 212 | 45.8 |
> | AutoMoE |  2.8  | 180  | 45.5 |
> | | 46% speedup | 15% speedup | |
>
>
> We add these results in Table 4 in our revision.
>
>
> References:
> * [Wang et al., 2020] [HAT](https://aclanthology.org/2020.acl-main.686/) ACL.
> * [So et al., 2022] [Primer](https://arxiv.org/abs/2109.08668) NeurIPS.
> * [Gao et al., 2022] [AutoBERT-Zero](https://ojs.aaai.org/index.php/AAAI/article/download/21311/version/19598/21060) AAAI.

---

### Official Review · Reviewer_vutH · 2022-10-23

**Confidence:** 5
**Clarity, Quality, Novelty And Reproducibility:** The paper is well-written. However, t…
**Correctness:** 3
**Technical Novelty And Significance:** 2
**Empirical Novelty And Significance:** 2
**Recommendation:** 3

**Strength And Weaknesses:**

Strengths:
1. This paper develops a new NAS approach by incorporating MoE module into the search space
2. This paper provides sufficient details of the proposed search space as well as interesting analysis.


Weaknesses:

1. Compared with HAT, the novelty of this paper seems very limited. If I understand correctly, the essential difference from HAT is including the MoE module into the search space. Besides this, all the others (including supernet training, evolutionary search algorithm) remain exactly the same as HAT.

2. How do the authors evaluate latency in Table 1? If the latency is measured based on a single sample, the practical latency for a mini-batch may be very slow since different sub-networks are activated for different inputs. Thus, it would be better to provide the comparisons of latency on both single sample and a mini-batch.

3. The performance improvement over HAT seems very marginal in Table 4.

4. In NAS papers, random search is always a strong baseline. Thus, it would be better to include the comparisons with random search in Table 4.

5. In Table 5, this paper only shows the results of the proposed method. It is interesting to see the comparisons between the proposed method and HAT, which is also able to produce architectures under specific constraints.

6. Does AutoMoE also need to train a latency predictor like HAT? If not, how to efficiently obtain the latency for candidate architectures during the search process?


**Summary Of The Paper:**

This paper focuses on neural machine translation and develops a new NAS approach. Specifically, the authors propose a transformer-based search space by incorporating the MoE module into it. Nevertheless, the novelty seems very limited since the proposed method can be regarded as a direct application of HAT on MoE architectures. Moreover, the performance improvement over HAT is marginal.

**Summary Of The Review:**

The novelty is limited since the proposed method can be regarded as a direct application of HAT on MoE architectures. Moreover, the performance improvement over HAT is marginal.
Maybe I missed something important. If there are some other essential differences, I am happy to change the rating.

---

> ### Author Response · Authors · 2022-11-12
> **Response to reviewer vutH**
>
> Thank you for your time to review our work and provide feedback. We respond to quoted comments below:
>
> * *“Compared with HAT, the novelty of this paper seems very limited. ... Besides this, all the others (including supernet training, evolutionary search algorithm) remain exactly the same as HAT.”*
>
> We **respectfully disagree** with this comment. Please see General Response, (1) and (2) for an explanation of the novelty of our work. See General Response, (3) for the major differences between HAT and AutoMoE.
> **To summarize:**
>    1. AutoMoE is the first work on NAS for MoE
>    2. Introduces the notion of adaptive computation with heterogeneous experts that is not explored in prior MoE works. This shortcoming has been posed as a future research direction in several recent works (MoE survey - Fedus et al., 2022; ST-MoE - Zoph et al., 2022).
>    3. AutoMoE incorporates MoE search space over HAT (which adopted the same search space as Transformers). This requires non-trivial sampling and routing designs (Figure 2); latency evaluators due to the stochastic nature of routing (Section 3.3)
>    4. Scalability as our MoE SuperNets are E times larger than HAT SuperNet (e.g., E=16 for 16 experts)
>    5. Significant speedup over HAT (23% FLOPs and 10% latency reduction over optimal HAT architectures)
>
> * *“How do the authors evaluate latency in Table 1? ... Thus, it would be better to provide the comparisons of latency on both single sample and a mini-batch.”*
>
> See General Response, (4b) for the evaluation setting. We adopt the same evaluation setting and script as HAT for a fair comparison. The latency is measured based on a single sample, following HAT.
>
> * *“The performance improvement over HAT seems very marginal in Table 4.”*
>
> AutoMoE obtains 23% FLOPs and 10% latency reduction over optimal HAT architectures while maintaining similar BLEU scores. By design, AutoMoE focuses on optimizing FLOPs and latency, while targeting a similar BLEU score as HAT. We further add latency evaluation to that reported in Table 4 as follows.
>
> **WMT’14 En-De**:
> |  Framework  | FLOPs (G) | Latency (ms) | BLEU |
> | ------------- |:-------------:|:-------------:|:-------------:|
> | HAT     | 3.5 | 205 | 28.2 |
> | AutoMoE |  2.9 | 190 | 28.2 |
> | | 17% speedup | 7.9% speedup |
>
> **WMT’14 En-Fr**:
> |  Framework  | FLOPs (G) | Latency (ms) | BLEU |
> | ------------- |:-------------:|:-------------:|:-------------:|
> | HAT     | 3.6 | 212 | 41.5 |
> | AutoMoE |  2.9 | 196 | 41.6 |
> | | 19% speedup | 7.5% speedup | |
>
> **WMT’19 En-De**:
> |  Framework  | FLOPs (G) | Latency (ms) | BLEU |
> | ------------- |:-------------:|:-------------:|:-------------:|
> | HAT     | 4.1 | 212 | 45.8 |
> | AutoMoE |  2.8  | 180  | 45.5 |
> | | 46% speedup | 15% speedup | |
>
> We include these results in Table 4 in our revision.
>
> * *"“In NAS papers, random search is always a strong baseline. Thus, it would be better to include the comparisons with random search in Table 4.”"*
> Thanks for the suggestion. Random search yields architectures with poorer BLEU score as shown in the following table:
>
> **WMT’14 En-De**:
> |  Framework  | #Active Params (M) | FLOPs (G) | Latency (ms) | BLEU |
> | ------------- |:-------------:|:-------------:|:-------------:|:-------------:|
> | Random Search     | 54 | 2.2 | 145 | 27.3 |
> | AutoMoE |  45 | 2.9 | 190 | 28.2 |
>
> We include these results in Table 3 in our revision.
>
> * *“In Table 5, this paper only shows the results of the proposed method. It is interesting to see the comparisons between the proposed method and HAT, which is also able to produce architectures under specific constraints.”*
>
> **Given the same latency constraint of 200 ms** (used in Table 4 for all NAS methods and baselines including HAT and AutoMoE), **we observe that AutoMoE-optimal architectures improve upon HAT-optimal architectures by 23% in FLOPs and 10% in latency reduction with matching BLEU scores.** We observe (from Table 9) latency based constraints to work better than FLOPs based constraints. Hence, comparisons in the paper have been made only with the same latency constraint.
>
> * *"“Does AutoMoE also need to train a latency predictor like HAT? If not, how to efficiently obtain the latency for candidate architectures during the search process?”"*
>
> Latency predictors (like the one used in HAT) are quite noisy for stochastic routing models like MoE’s and exhibited big variance and MSE in our experiments. Accordingly, we adopt “partially gold” latency where we directly measure the latency on the target device for a small number of forward passes and compute the truncated mean. We discuss these design choices in Section 3.3. Table 3 reports the end-to-end training cost (GPU hours) of each method including the latency measurement.

---

### Official Review · Reviewer_Yxc3 · 2022-10-24

**Confidence:** 4
**Clarity, Quality, Novelty And Reproducibility:** This paper has good clarity, fair qua…
**Correctness:** 3
**Technical Novelty And Significance:** 2
**Empirical Novelty And Significance:** 2
**Recommendation:** 5

**Strength And Weaknesses:**

---
Strengths:
- The paper is very well-written and easy to follow.
- The proposed framework is technically sound and performs well on machine translation benchmarks.
- The paper provides sufficient implementation details (e.g., design space configuration, and search and training hyperparameters). The submission also comes with a code implementation, which will facilitate other researchers' reproduction.

---
Weaknesses:
- The technical novelty of this paper is a bit limited. The core contribution lies in the search space design since the other components (i.e., super network training and evolutionary search) are standard (used in most neural architecture search papers). However, the authors adopt most of their search space designs from HAT except those related to MoE. The support for elastic MoE also seems straightforward (basically the same as FFN). Though I like the idea of incorporating MoE into the transformer's design space, I am not confident whether these technical contributions are sufficient for publication.
- The experimental evaluation needs improvement:
  - Despite mentioning the baseline with homogeneous experts in the text, the authors have not provided the corresponding results in Table 4.
  - The authors have only presented #FLOPs in Table 4. However, #FLOPs reduction does not necessarily translate into measured speedup.
  - The authors have compared AutoMoE with manually-designed baselines in Table 1. However, I suspect the shallower encoder contributes the most to the latency and #FLOPs reduction.

---

**Summary Of The Paper:**

This paper studies efficient sparsely activated transformers. The authors expand the search space of transformers with a few mixture-of-experts (MoE) design choices: i.e., the number and size of experts after each transformer block. Then, the authors apply weight-sharing super network training and evolutionary search to explore the best sparsely activated MoE architecture under the given resource constraints. The proposed AutoMoE has achieved moderate #FLOPs reduction compared with the dense transformer.

**Summary Of The Review:**

My current recommendation is primarily based on the limited novelty of this paper. However, I would love to see the authors' response before making the final recommendation.

---

> ### Author Response · Authors · 2022-11-12
> **Response to reviewer Yxc3**
>
> Thank you for your time to review our work and provide feedback. We respond to quoted comments below:
>
> * *“The technical novelty of this paper is a bit limited. The core contribution lies in the search space design ... Though I like the idea of incorporating MoE into the transformer's design space, I am not confident whether these technical contributions are sufficient for publication.”*
>
> Our “General Response” comprehensively addresses several points including novelty: please see (1) and (2). **To summarize**:
>   1. AutoMoE is the first work on NAS for MoE.
>    2. Introduces the notion of adaptive computation with heterogeneous experts that is not explored in prior MoE works. This shortcoming has
> been posed as a future research direction in several recent works (MoE survey - Fedus et al., 2022a; ST-MoE - Zoph et al., 2022).
>    3. AutoMoE incorporates MoE search space over HAT (which adopted the same search space as Transformers). This requires non-trivial sampling and routing designs (Figure 2); latency evaluators due to the stochastic nature of routing (Section 3.3)
>    4. Scalability as our MoE SuperNets are E times larger than HAT SuperNet (e.g., E=16 for 16 experts)
>    5. Significant speedup over HAT (23% FLOPs and 10% latency reduction over optimal HAT architectures)
>
> * *“Despite mentioning the baseline with homogeneous experts in the text, the authors have not provided the corresponding results in Table 4.”*
>
> The results for the baseline with homogeneous experts are given in Table 1 (manually designed) as well as Table 6 (ablation results) in the submitted version. We will add these results to Table 4 in our revision.
>
> * *“The authors have only presented #FLOPs in Table 4. However, #FLOPs reduction does not necessarily translate into measured speedup.”*
>
> Please note that based on our experiments, AutoMoE generated architectures achieve better latency in addition to better FLOPs as shown in the following tables:
>
> **WMT’14 En-De**:
> |  Framework  | FLOPs (G) | Latency (ms) | BLEU |
> | ------------- |:-------------:|:-------------:|:-------------:|
> | HAT     | 3.5 | 205 | 28.2 |
> | AutoMoE |  2.9 | 190 | 28.2 |
> | | 17% speedup | 7.9% speedup |
>
> **WMT’14 En-Fr**:
> |  Framework  | FLOPs (G) | Latency (ms) | BLEU |
> | ------------- |:-------------:|:-------------:|:-------------:|
> | HAT     | 3.6 | 212 | 41.5 |
> | AutoMoE |  2.9 | 196 | 41.6 |
> | | 19% speedup | 7.5% speedup | |
>
> **WMT’19 En-De**:
> |  Framework  | FLOPs (G) | Latency (ms) | BLEU |
> | ------------- |:-------------:|:-------------:|:-------------:|
> | HAT     | 4.1 | 212 | 45.8 |
> | AutoMoE |  2.8  | 180  | 45.5 |
> | | 46% speedup | 15% speedup | |
>
> We add these results to Table 4 in our revision.
>
> * *“The authors have compared AutoMoE with manually-designed baselines in Table 1. However, I suspect the shallower encoder contributes the most to the latency and #FLOPs reduction.”*
>
> All models in the paper including HAT and AutoMoE use 6 encoder layers. From Figure 3e, we observe decoder latency to contribute more than 90% of the end-to-end latency for autoregressive decoding. Therefore, all NAS works, including HAT focus on reducing decoder layers as opposed to encoder layers. We adopt the same evaluation setting as HAT.
>
>
> References:
> * [Fedus et al., 2022a] [A Review of Sparse Expert Models in Deep Learning.](https://arxiv.org/abs/2209.01667) arXiv.
> * [Zoph et al., 2022] [ST-MoE](https://arxiv.org/abs/2202.08906) arXiv.

---

> > ### Comment · Reviewer_Yxc3 · 2022-11-17
> > **Reviewer Response**
> >
> > Thank you so much for your response! I am afraid I have to disagree with your argument on technical novelty.
> > - **Search space.** When adding a design choice to the search space, it is a common practice to make it elastic/heterogeneous: e.g., kernel size and expansion ratio. Adding MoE into the design space is a reasonable engineering effort but cannot be considered a solid technical contribution.
> > - **Super network training.** As mentioned in the main review, the support for elastic MoE is basically the same as FFN, which can't be considered novel.
> > - **Latency measurement.** If this is a technical contribution, the authors should provide experimental results showing that the latency predictor does not work well. Nevertheless, the proposed "partially gold latency" is not new either. It is a fairly standard approach to reducing the variance of latency measurement.
> >
> > Besides, the added latency measurement also confirms my concern. The latency reduction is relatively incremental compared with the baseline. Therefore, I will keep my score based on the current discussion.

---

### Author Response · Authors · 2022-11-12
**General Response to all Reviewers (i)**


We thank all reviewers for their insightful comments. We want to highlight the following aspects of our work:

1. **Problem Novelty:** Ours is the **first work** to explore sparsely activated Transformers (e.g., MoE’s) in the NAS search space. The main contribution of the work is to design the sparse search space and demonstrate that it works better than the traditional dense search space for NAS. Although intuitive, this has never been shown before.

2. **Technical Novelty:**
   - **Research builds over prior work.** HAT (Wang et al., 2020) did not introduce SuperNet training or evolutionary search. Instead, it demonstrated for the first time that the end-to-end framework works well for NMT tasks. Similarly, BERT [Devlin et al., 2019] did not introduce Transformers; or the pre-train/fine-tune paradigm which existed before in ELMo (Peters et al., 2018). Instead, it demonstrated that the end-to-end combination achieves SOTA for NLU tasks. In the same way, AutoMoE introduces an end-to-end framework for bringing the MoE search space into NAS. This has never been explored before.
   - **Adaptive computation.** Prior MoE works (e.g., GShard (Lepikhin et al., 2021); Switch Transformer (Fedus et al., 2022b); ST-MoE (Zoph et al., 2022)) are non-adaptive i.e. they perform the same computation for every input due to the fixed size of the expert FFN’s. **AutoMoE is the first work to support adaptive computation** due to heterogeneous experts, where input tokens are routed to experts of different sizes. This shortcoming has been posed as a future research direction in several recent works (Fedus et al., 2022a; Zoph et al., 2022).

3. **Differences with HAT and MoE’s:**
   - **Search space:** HAT search space is not different from traditional Transformers. AutoMoE search space incorporates MoE’s in the FFN’s, but allows them to be heterogeneous. This is a significant advance from all existing MoE architectures (e.g., GShard, SwitchTransformer, ST-MoE) which assume homogeneity. AutoMoE allows different numbers of experts to be placed in different layers, of variable sizes, and even to skip them. In the process, we develop design principles for efficient MoE’s (Section 5.2) which addresses a longstanding open challenge (see Fedus et al., 2022a; Zoph et al., 2022).
   - **Non-trivial extension of HAT SuperNet training:** There is no notion of “routing” in HAT. By contrast, the key challenges in AutoMoE include — introducing “routers”, “FFN experts” of dynamic sizes, choices to introduce them in encoder vs. decoder, choices to skip them to save computation, variable numbers, etc. These need to be integrated with the subnetwork sampling, which inherits partial weights from the SuperNet. These require non-trivial engineering and implementation (detailed in Section 3.2, Figure 2) particularly for scaling. In contrast to the biggest SuperNet in HAT, AutoMoE supernet is “E” times larger (e.g., E=16 for 16 experts). We have submitted the code and implementation details for reproducibility.
   - **Non-trivial extensions for evolutionary search:** Given the stochastic nature of routing, traditional latency predictors incur large errors and cannot be used reliably for MoE’s. AutoMoE introduces “partially gold latency” for more accurate estimation (see Section 3.3). Unlike HAT, AutoMoE does not require creation of latency datasets and training of latency predictors for each hardware, which can save overall search time. AutoMoE also uses only half of the evolutionary search iterations (15 out of 30) compared to HAT.

4. **Evaluation and Significant Performance Improvement over HAT**
   - **AutoMoE reduces FLOPs by 23% (Table 4) and latency by 10% (table in individual response) over optimal HAT architectures under the same latency constraint (200 ms)** in the aggregate, across benchmark NMT tasks, while maintaining similar task performance.
   - We use the **same evaluation setting and code for FLOPs and latency computation as HAT.** All the models in Table 4 use **same number of encoder layers**, including HAT and AutoMoE (note that FLOPs/latency reduction comes from reduced decoder layers, not encoder; decoders incur more than 90% of the end-to-end latency for autoregressive decoding - Figure 3e)
   - Further results (e.g., homogeneous vs. heterogeneous) are provided as ablation in Table 6.

References:
* [Wang et al., 2020] [HAT](https://aclanthology.org/2020.acl-main.686/) ACL.
* [Devlin et al., 2019] [BERT](https://aclanthology.org/N19-1423/) NAACL.
* [Peters et al., 2018] [ELMo](https://aclanthology.org/N18-1202/) NAACL.
* [Fedus et al., 2022a] [A Review of Sparse Expert Models in Deep Learning.](https://arxiv.org/abs/2209.01667) arXiv.
* [Zoph et al., 2022] [ST-MoE](https://arxiv.org/abs/2202.08906) arXiv.
* [Fedus et al., 2022b] [Switch Transformers](https://www.jmlr.org/papers/v23/21-0998.html) JMLR.
* [Lepikhin et al., 2021] [GShard](https://openreview.net/forum?id=qrwe7XHTmYb) ICLR.

---

### Author Response · Authors · 2022-11-16
**Acknowledging our response**

Dear Reviewers,

We would be thankful if you can acknowledge our response, or let us know if there are any comments that we can address before the end of the author response period. Please consider revising your rating if we are able to address any of your major concerns.

---

### Author Response · Authors · 2022-11-19
**Uploaded the revision**

Dear Reviewers,

We have uploaded a revised version with your recommended changes (highlighted in “red” color in the revision):
1. A paragraph that explicitly explains the novelty of our work in Section 1 (all reviewers).
2. Added manual MoE results (Transformer Sparse) in Table 3 (reviewer Yxc3).
3. Added random search results in Table 3 (reviewer vutH).
4. Added latency results in Table 4 and mentioned the improvements in abstract and introduction (all reviewers).
5. Added the full architecture diagram of AutoMoE generated architectures in Figure 4, 5, 6.
6. Added the latency improvement with the proposed partially gold latency vs. latency predictor in Table 8 (reviewer Yxc3).

Thanks.

---

### Decision · Program_Chairs · 2023-01-20

**Decision:**

Reject

**Justification For Why Not Higher Score:**

Limited contributions.

**Justification For Why Not Lower Score:**

N/A

**Metareview: Summary, Strengths And Weaknesses:**

All reviewers find the contributions in the paper limited and straightforward application of existing methods. Further the improvements obtained by the costly NAS search seem marginal over standard baselines. Author's response didn't change reviewers mind and I also agree with reviewers that the current submission looks like a straightforward application of existing results. Overall the current submission is weak and will need a significant revision before resubmission. Hence I recommend rejection.